# Private Set Generation with Discriminative Information

**Dingfan Chen    Raouf Kerkouche    Mario Fritz**
CISPA Helmholtz Center for Information Security
`{dingfan.chen, raouf.kerkouche, fritz}@cispa.de`

## Abstract

Differentially private data generation techniques have become a promising solution to the data privacy challenge — it enables sharing of data while complying with rigorous privacy guarantees, which is essential for scientific progress in sensitive domains. Unfortunately, restricted by the inherent complexity of modeling high-dimensional distributions, existing private generative models are struggling with the utility of synthetic samples. In contrast to existing works that aim at fitting the complete data distribution, we directly optimize for a small set of samples that are representative of the distribution under the supervision of discriminative information from downstream tasks, which is generally an easier task and more suitable for private training. Our work provides an alternative view for differentially private generation of high-dimensional data and introduces a simple yet effective method that greatly improves the sample utility of state-of-the-art approaches.

## 1    Introduction

Data sharing is vital for the growth of machine learning applications in numerous domains. However, in many application scenarios, data sharing is prohibited due to the private nature of data (e.g., individual data stored on mobile devices, medical treatments, and banking records) and the corresponding stringent regulations, which greatly hinders technological progress. Differentially private (DP) data publishing [8, 9, 11] provides a compelling solution to such challenge, where only a sanitized form of the data is publicly released. Such sanitized synthetic data can be leveraged as if it were the real data, analyzed with established toolchains, and can be shared openly to the public, facilitating technological advance and reproducible research in sensitive domains.

Yet, generation of high-dimensional data with DP guarantees is highly challenging and traditional DP algorithms designed for capturing low-dimensional statistical characteristics are not applicable to this task [28, 13, 4, 35]). Instead, inspired by the great successes of deep generative models in learning high-dimensional representations, recent works [5, 6, 40, 41, 2, 12] adopt deep generative neural networks as the underlying generation backbone and incorporate the privacy constraints into the training procedure, such that any privacy leakage upon disclosing the data generator is bounded.

However, these methods have common shortcomings: *(i)* deep generative models are known to be data-demanding [16], which becomes even harder to train when considering the privacy constraints [6, 5]; *(ii)* they do not guarantee any optimal solution for the downstream task (e.g. classification). In fact, existing models are still struggling to generate sanitized data that is useful for downstream data analysis tasks. For example, when training a convolutional neural network (ConvNet) classifier on the private generated data, the highest test accuracy reported in literature is $< 85\%$ for MNIST dataset with $(\varepsilon, \delta) = (10, 10^{-5})$ [5], which lags far behind the discriminative baseline ($> 98\%$ with $(\varepsilon, \delta) = (1.2, 10^{-5})$ [32]) and makes private generative models less appealing for many practical scenarios with data analysis as the end goal.

36th Conference on Neural Information Processing Systems (NeurIPS 2022).

In this work, we learn to synthesize informative samples that are privacy-preserving and are optimized to train neural networks for downstream tasks. In contrast to existing approaches, we directly optimize a small set of samples instead of the deep generative models that is notoriously difficult to train in a private manner. Moreover, we exploit discriminative information from downstream tasks to guide the samples towards containing more useful information for downstream analysis. Compared to existing works, we improve the task utility by a large extent (up to 10% downstream test accuracy improvement over state-of-the-art approach), while still preserving the flexibility and generality across varying configurations for downstream analysis. As an added benefit, our formulation naturally distilled the knowledge of original data into a much smaller set, which largely saves the memory and computational consumption for downstream analysis.

We summarize our main contributions as follows.

- We present a new perspective of private high-dimensional data generation, with which we aim to bridge the utility and generality gap between the private generative and discriminative models. We believe this alternative view opens up new possibilities in different research directions ranging from private analysis to generation.
- We introduce a simple yet effective method for generating informative samples that are optimized for training downstream neural networks, while maintaining generality as well as reducing the memory and computation consumption as added benefits.
- Experimental results demonstrate that, in comparison to existing works, our work improves the sample utility by a large margin and offers superior practicability for real-world application scenarios.

## 2  Related Work

**Differentially Private Generative Models**    Training deep generative models in a private manner has become the default choice for private high-dimensional data generation. Existing methods typically adopt differentially private stochastic gradient descent (DP-SGD) [1, 31, 6, 5] or Private Aggregation of Teacher Ensembles (PATE) [23, 24, 20, 36] to equip the deep generative models with rigorous privacy guarantees. Despite significant progress in mitigating training instabilities and improving generation (visual) quality, existing works are still far from being optimal in terms of the sample utility. This is mainly because existing works are attempting to solve a problem that is inherently hard and almost impossible to be solved accurately under the current private training framework. In contrast, we directly optimize the samples (rather than the deep generative models that are much harder to train in a private setting) and exploit the knowledge from a general class of downstream tasks that can be employed on the samples to further guide the training.

**Coreset Selection and Generation**    Our work is largely motivated by recent success in distilling a large dataset into a much smaller set of representative samples, i.e., the coreset. For example, samples from a dataset are selected to be representative based on their ability to mimic the gradient signal [22], hardness to fit [30], distance to the cluster centers [38, 27], etc. Instead of selecting samples from the dataset, our work focus on synthesizing informative samples from scratch [37, 43, 42, 19] under DP constraints, and optimizing the sample utility for training downstream neural networks. While recent work [7] has shown promising results in dataset distillation under privacy concerns, obtaining strict privacy guarantees has remained challenging. Our set generation formulation is also similar in spirit to works in the field of private queries release [28, 13, 4, 14] which synthesize a set of pseudo-data (under DP guarantees) that is representative of the original data in answering linear queries. However, as neural networks exhibit highly nonlinear properties, methods targeted at linear queries are generally not applicable to our case and are algorithmically distinct from approaches designed for neural nets.

## 3  Background

We consider the standard central model of DP in this paper. We below present several definitions and theorems that will be used in this work.

**Definition 3.1.** (Differential Privacy (DP) [8]) A randomized mechanism $\mathcal{M}$ with range $\mathcal{R}$ is $(\varepsilon, \delta)$-DP, if

$$Pr[\mathcal{M}(\mathcal{D}) \in \mathcal{O}] \leq e^{\varepsilon} \cdot Pr[\mathcal{M}(\mathcal{D}') \in \mathcal{O}] + \delta \tag{1}$$

holds for any subset of outputs $\mathcal{O} \subseteq \mathcal{R}$ and for any adjacent datasets $\mathcal{D}$ and $\mathcal{D}'$, where $\mathcal{D}$ and $\mathcal{D}'$ differ from each other with only one training example, i.e., $\mathcal{D}' = \mathcal{D} \cup \{x\}$ for some $x$ (or vice versa). $\mathcal{M}$ corresponds to the set generation algorithm in our case, $\varepsilon$ is the upper bound of privacy loss, and $\delta$ is the probability of breaching DP constraints. DP guarantees the difficulty of inferring the presence of an individual in the private dataset by observing the generated set of samples $\mathcal{M}(\mathcal{D})$.

Our approach is built on top of the Gaussian mechanism defined as follows.

**Definition 3.2.** (Gaussian Mechanism [10]) Let $f : X \to \mathbb{R}^d$ be an arbitrary $d$-dimensional function with sensitivity being

$$\Delta_2 f = \max_{\mathcal{D},\mathcal{D}'} \|f(\mathcal{D}) - f(\mathcal{D}')\|_2 \tag{2}$$

over all adjacent datasets $\mathcal{D}$ and $\mathcal{D}'$. The Gaussian Mechanism $\mathcal{M}_\sigma$, parameterized by $\sigma$, adds noise into the output, i.e.,

$$\mathcal{M}_\sigma(x) = f(x) + \mathcal{N}(0, \sigma^2 I). \tag{3}$$

$\mathcal{M}_\sigma$ is $(\varepsilon, \delta)$-DP for $\sigma \geq \sqrt{2 \ln (1.25/\delta)} \Delta_2 f / \varepsilon$.

Any privacy cost is bounded upon releasing the private set of generated data due to the closedness of DP under post-processing.

**Theorem 3.1.** (Post-processing [10]) If $\mathcal{M}$ satisfies $(\varepsilon, \delta)$-DP, $F \circ \mathcal{M}$ will satisfy $(\varepsilon, \delta)$-DP for any data-independent function $F$ with $\circ$ denoting the composition operator.

# 4 Method

We consider a standard classification task where we are given a private dataset $\mathcal{D} = \{(\boldsymbol{x}_i, y_i)\}_{i=1}^{N}$ with $\boldsymbol{x}_i \in \mathbb{R}^d$ the feature, $y_i \in \{1, ..., L\}$ the class label, $N$ the number of samples, $L$ the number of label classes. Our objective is to synthesize a set of samples $\mathcal{S} = \{(\boldsymbol{x}_i^{\mathcal{S}}, y_i^{\mathcal{S}})\}_{i=1}^{M}$ such that (i) samples in $\mathcal{S}$ have the same form as data in $\mathcal{D}$; (ii) a neural network trained on $\mathcal{S}$ should maximally match generalization performance of a deep neural network that is trained on $\mathcal{D}$; (iii) the privacy leakage of $\mathcal{D}$ when releasing $\mathcal{S}$ is upper bounded by a pre-defined privacy level $(\varepsilon, \delta)$.

Let $F(\cdot\,;\boldsymbol{\theta}^{\mathcal{D}})$ and $F(\cdot\,;\boldsymbol{\theta}^{\mathcal{S}})$ be the deep neural networks parameterized by $\boldsymbol{\theta}^{\mathcal{D}}$ and $\boldsymbol{\theta}^{\mathcal{S}}$ that are trained on $\mathcal{D}$ and $\mathcal{S}$ respectively. The objective can be formulated as:

$$\mathbb{E}_{(\boldsymbol{x},y) \sim P_{\mathcal{D}}}[\ell(F(\boldsymbol{x};\boldsymbol{\theta}^{\mathcal{D}}), y)] \simeq \mathbb{E}_{(\boldsymbol{x},y) \sim P_{\mathcal{D}}}[\ell(F(\boldsymbol{x};\boldsymbol{\theta}^{\mathcal{S}}), y)] \tag{4}$$

where $\ell$ denotes the loss function (e.g., cross-entropy for the classification task) and the expectation is taken over the real data distribution $P_{\mathcal{D}}$.

Equation 4 can be naturally achieved once $\boldsymbol{\theta}^{\mathcal{S}} \approx \boldsymbol{\theta}^{\mathcal{D}}$. In particular, when given the same initialization $\boldsymbol{\theta}_0^{\mathcal{D}} = \boldsymbol{\theta}_0^{\mathcal{S}}$, solving for $\boldsymbol{\theta}_t^{\mathcal{S}} \approx \boldsymbol{\theta}_t^{\mathcal{D}}$ at each training iteration $t$ leads to $\boldsymbol{\theta}^{\mathcal{S}} \approx \boldsymbol{\theta}^{\mathcal{D}}$ as desired. This can be achieved by optimizing the synthetic set $\mathcal{S}$ such that it yields a similar gradient as if the network is trained on the real dataset at each iteration $t$:

$$\min_{\mathcal{S}} \mathcal{L}_{\mathrm{dis}}(\nabla_{\boldsymbol{\theta}}\mathcal{L}(\mathcal{S}, \boldsymbol{\theta}_t), \nabla_{\boldsymbol{\theta}}\mathcal{L}(\mathcal{D}, \boldsymbol{\theta}_t)) \tag{5}$$

where $\nabla_{\boldsymbol{\theta}}\mathcal{L}(\mathcal{S}, \boldsymbol{\theta}_t))$ corresponds to the gradient of the classification loss on the synthetic set $\mathcal{S}$, $\nabla_{\boldsymbol{\theta}}\mathcal{L}(\mathcal{D}, \boldsymbol{\theta}_t)$ denotes the stochastic gradient on the real data, and $\mathcal{L}_{\mathrm{dis}}$ is a sum of cosine distances between the gradients at each layer [43, 42] (See supplementary material for more details).

To mimic the training procedure, $\mathcal{S}$ and the network $F(\cdot;\boldsymbol{\theta})$ are updated jointly in an iterative manner, where in each outer iteration the $\mathcal{S}$ is trained to minimize the gradient matching loss $\mathcal{L}_{\mathrm{dis}}$ and in each inner iterations the network parameters $\boldsymbol{\theta}_t$ are optimized towards minimizing the classification loss on the synthetic set $\mathcal{S}$. Moreover, $\mathcal{S}$ is optimized over multiple initializations of network parameters $\boldsymbol{\theta}_0$ to ensure the generalization ability of $\mathcal{S}$ over different random initialization when

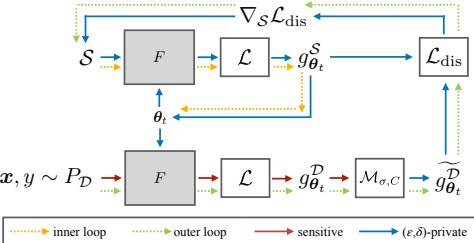

**Figure 1:** Illustration for the training pipeline.

training a downstream model. The objective can be summarize as follows [37, 43, 42]:

$$\mathcal{S} = \arg\min_{\mathcal{S}} \mathbb{E}_{\boldsymbol{\theta}_0 \sim P_{\boldsymbol{\theta}_0}} \sum_{t=0}^{T-1} [\mathcal{L}_{\text{dis}}(\nabla_{\boldsymbol{\theta}} \mathcal{L}(\mathcal{S}, \boldsymbol{\theta}_t), \nabla_{\boldsymbol{\theta}} \mathcal{L}(\mathcal{D}, \boldsymbol{\theta}_t))] \qquad (6)$$

where $P_{\boldsymbol{\theta}_0}$ stands for the distribution over the initialization of network parameters.

We incorporate DP constraints by sanitizing the stochastic gradient on real data $\nabla_{\boldsymbol{\theta}} \mathcal{L}(\mathcal{D}, \boldsymbol{\theta}_t)$ at each outer iteration, while leaving the inner iterations unchanged as their privacy is guaranteed by the post-processing theorem 3.1. The final objective can be formulated as follows:

$$\mathcal{S} = \arg\min_{\mathcal{S}} \mathbb{E}_{\boldsymbol{\theta}_0 \sim P_{\boldsymbol{\theta}_0}} \sum_{t=0}^{T-1} [\mathcal{L}_{\text{dis}}(g_{\boldsymbol{\theta}_t}^{\mathcal{S}}, \widetilde{g_{\boldsymbol{\theta}_t}^{\mathcal{D}}})] \qquad (7)$$

where we use $\widetilde{g_{\boldsymbol{\theta}_t}^{\mathcal{D}}}$ to denote the parameter gradient on $\mathcal{D}$ that is sanitized via Gaussian mechanism 3.2, and $g_{\boldsymbol{\theta}_t}^{\mathcal{S}}$ to denote the parameter gradient on $\mathcal{S}$. The whole pipeline is summarized in Algorithm 1. We use the subsampled Renyi-DP accountant [1, 21] to compute the overall privacy cost accumulated for iteratively updating $\mathcal{S}$. Note that the training procedure and the privacy computation are approximately as simple as training a classification network with DP-SGD, which in general has lower difficulty than training a DP deep generative models as done in existing works (witnessed by a significant performance gap in terms of the classification accuracy). Moreover, in contrast to previous works, our synthetic set $\mathcal{S}$ is directly optimized for downstream tasks, which naturally leads to superior downstream utility to existing approaches.

---

**Algorithm 1:** Private Set Generation (PSG)

---

**Input:** Dataset $\mathcal{D} = \{(\boldsymbol{x}_i, y_i)\}_{i=1}^{N}$, learning rate for update network parameters $\tau_{\boldsymbol{\theta}}$ and $\tau_{\mathcal{S}}$, batch size $B$, gradient clipping bound $C$, number of runs $R$, outer iterations $T$, inner iterations $J$, batches $K$, desired privacy cost $\varepsilon$ given a pre-defined $\delta$

**Output:** Synthetic set $\mathcal{S}$

Compute the required DP noise scale $\sigma$ numerically [1, 21] so that the privacy cost equals $\varepsilon$ after the training; Initialize synthetic set $\mathcal{S}$ (features $\boldsymbol{x}^{\mathcal{S}}$ are from Gaussian noise; labels are balanced set depending on the pre-defined number of samples per class) ;

**for** *run* **in** $\{1, ..., R\}$ **do**

    Initialize model parameter $\boldsymbol{\theta}_0 \sim P_{\boldsymbol{\theta}_0}$;

    **for** *outer_iter* **in** $\{1, ..., T\}$ **do**

        $\boldsymbol{\theta}_{t+1} = \boldsymbol{\theta}_t$

        **for** *batch_index* **in** $\{1, ..., K\}$ **do**

            Uniformly sample random batch $\{(\boldsymbol{x}_i, y_i)\}_{i=1}^{B}$ from $\mathcal{D}$;

            **for** *each* $(\boldsymbol{x}_i, y_i)$ **do**

                // Compute per-example gradients on real data

                $g_{\boldsymbol{\theta}_t}^{\mathcal{D}}(\boldsymbol{x}_i) = \ell(F(\boldsymbol{x}_i; \boldsymbol{\theta}_t), y_i)$

                // Clip gradients

                $\widetilde{g_{\boldsymbol{\theta}_t}^{\mathcal{D}}}(\boldsymbol{x}_i) = g_{\boldsymbol{\theta}_t}^{\mathcal{D}}(\boldsymbol{x}_i) \cdot \min(1, C/\|g_{\boldsymbol{\theta}_t}^{\mathcal{D}}(\boldsymbol{x}_i)\|_2)$

            **end**

            // Add noise to average gradient with Gaussian mechanism

            $\widetilde{g_{\boldsymbol{\theta}_t}^{\mathcal{D}}} = \frac{1}{B} \sum_{i=1}^{B} (\widetilde{g_{\boldsymbol{\theta}_t}^{\mathcal{D}}}(\boldsymbol{x}_i) + \mathcal{N}(0, \sigma^2 C^2 I))$

            // Compute parameter gradients on synthetic data and update $\mathcal{S}$

            $g_{\boldsymbol{\theta}_t}^{\mathcal{S}} = \nabla_{\boldsymbol{\theta}} \mathcal{L}(\mathcal{S}, \boldsymbol{\theta}_t)) = \frac{1}{M} \sum_{i=1}^{M} \ell(F(\boldsymbol{x}_i^{\mathcal{S}}; \boldsymbol{\theta}_t), y_i^{\mathcal{S}})$

            $\mathcal{S} = \mathcal{S} - \tau_{\mathcal{S}} \cdot \nabla_{\mathcal{S}} \mathcal{L}_{\text{dis}}(g_{\boldsymbol{\theta}_t}^{\mathcal{S}}, \widetilde{g_{\boldsymbol{\theta}_t}^{\mathcal{D}}})$

        **end**

    **end**

    **for** *inner_iter* **in** $\{1, ..., J\}$ **do**

        // Update network parameter using $\mathcal{S}$

        $\boldsymbol{\theta}_t = \boldsymbol{\theta}_t - \tau_{\boldsymbol{\theta}} \cdot \nabla_{\boldsymbol{\theta}} \mathcal{L}(\mathcal{S}, \boldsymbol{\theta}_t)$

    **end**

**end**

**return** Synthetic set $\mathcal{S}$

---

| | (a) | | | | | (b) | | | | | |
|---|---|---|---|---|---|---|---|---|---|---|---|
| | MNIST | | FashionMNIST | | | MNIST | | | FashionMNIST | | |
| | $\varepsilon$=1 | $\varepsilon$=10 | $\varepsilon$=1 | $\varepsilon$=10 | | spc=10 | spc=20 | full | spc=10 | spc=20 | full |
| DP-CGAN | - | 52.5 | - | 50.2 | Real | 93.6 | 95.9 | 99.6 | 74.4 | 77.4 | 93.5 |
| G-PATE | 58.8 | 80.9 | 58.1 | 69.3 | DPSGD | - | - | 96.5 | - | - | 82.9 |
| DataLens | 71.2 | 80.7 | 64.8 | 70.6 | DP-CGAN | 57.4 | 57.1 | 52.5 | 51.4 | 53.0 | 50.2 |
| GS-WGAN | - | 84.9 | - | 63.1 | GS-WGAN | 83.3 | 85.5 | 84.9 | 58.7 | 59.5 | 63.1 |
| DP-Merf | 72.7 | 85.7 | 61.2 | 72.4 | DP-Merf | 80.2 | 83.2 | 85.7 | 66.6 | 67.9 | 72.4 |
| DP-Sinkhorn | - | 83.2 | - | 71.1 | Ours | **94.9** | **95.6** | - | **75.6** | **77.7** | - |
| Ours (spc=20) | **80.9** | **95.6** | **70.2** | **77.7** | | | | | | | |

**Table 1:** Test accuracy (%) on real data of downstream ConvNet classifiers when training on the synthetic set with $\delta = 10^{-5}$. **(a)** Comparison under different privacy cost $\varepsilon \in \{1, 10\}$. **(b)** Comparison when varying the number of samples per class (spc) for training the downstream ConvNet with $\varepsilon = 10$, while "full" corresponds to 6000 samples per class. We show the results when training on real data non-privately and with DPSGD [1] as reference.

## 5 Experiment

### 5.1 Classification

We first compare private set generation (PSG) with existing DP generative models on standard classification benchmarks including MNIST [17] and FashionMNIST [39].

**Setup.** We use by default a ConvNet with 3 blocks where each block contains one Conv layer with 128 filters, followed by Instance Normalization [33], ReLU activation and AvgPooling modules, and a fully connected (FC) layer as the final output layer. We initialize the network parameters using Kaiming initialization [15] and the synthetic samples using standard Gaussian. We report the averaged results over 3 runs of experiments for all the comparisons. We list below the default hyperparameters used for the main experiments and refer to the supplementary material for more details: Clipping bound $C = 0.1$, $R =$1000 for $\varepsilon = 10$ (and 200 for $\varepsilon = 1$), number of samples per class (spc) $\in \{10, 20\}$, $K = 10$, $T = 10$ for spc=10 (and =20 for spc=20).

**Comparison to state of the art.** We show in Table 1a the results of, to the best of our knowledge, all existing DP high-dimensional data generation methods (whose validity has been justified via peer review at top-tier conferences) that report results on the benchmark datasets we consider. These include DP-CGAN [31], G-PATE [20], DataLens [36], GS-WGAN [6], DP-Merf [12], DP-Sinkhorn [5]. For methods that are not open-sourced, we report the original results from the published paper. As shown in Table 1a, our formulation results in significant improvement in the sample utility (measured by test accuracy on real data) for training downstream classification models. Specifically, the improvement is consistent and significant (around 5-10% increase over different configurations) for both the low privacy budget regime ($\varepsilon$=1) (around 8-9% improvement over SOTA in this case) and a relatively high privacy regime ($\varepsilon$=10) where all the investigated methods achieve convergence (around 10% and 5% increase in test accuracy for MNIST and FashionMNIST, respectively). Note that in contrast to most existing methods that show superiority only for a certain range of privacy levels, our improvement covers a wide range, if not all, of practical scenarios spanning across different privacy levels.

We then focus on the open-sourced methods that are strictly comparable (e.g., G-PATE and DataLens provide data-dependent $\varepsilon$, i.e., publishing $\varepsilon$ value will introduce privacy cost and are thus not directly comparable) to ours and conduct a comprehensive investigation through different angles.

**Memory and computation cost.** We additionally show that our method is the only one that simultaneously shows advantages in reducing the memory and computation consumption of downstream analysis. As shown in Table 1b, training the classifier with full (6000 samples per class) size of samples in most cases yields an upper bound for the test accuracy, while training on randomly subsampled smaller sets will decrease the performance, unless the generated samples are not informative such that they can be harmful to the downstream tasks (e.g., for DP-CGAN). In contrast, we directly optimize

| | MNIST | | | | | | FashionMNIST | | | | | |
|---|---|---|---|---|---|---|---|---|---|---|---|---|
| | ConvNet | LeNet | AlexNet | VGG11 | ResNet18 | MLP | ConvNet | LeNet | AlexNet | VGG11 | ResNet18 | MLP |
| Real | 99.6 | 99.2 | 99.5 | 99.6 | 99.7 | 98.3 | 93.5 | 88.9 | 91.5 | 93.8 | 94.5 | 86.9 |
| DP-CGAN | 50.2 | 52.6 | 52.1 | 54.7 | 51.8 | 54.3 | 50.2 | 52.6 | 52.1 | 54.7 | 51.8 | 54.3 |
| GS-WGAN | 84.9 | 83.2 | 80.5 | 87.9 | 89.3 | 74.7 | 54.7 | 62.7 | 55.1 | 57.3 | 58.9 | 65.4 |
| DP-Merf | 85.7 | 87.2 | 84.4 | 81.7 | 81.3 | 85.0 | 72.4 | 67.9 | 64.9 | 70.1 | 66.7 | **73.1** |
| Ours (spc=10) | 94.9 | 91.3 | 90.3 | 93.6 | **94.3** | 86.1 | 75.6 | **68.0** | **66.2** | 74.7 | **72.1** | 62.8 |
| Ours (spc=20) | **95.6** | **93.0** | **92.3** | **94.5** | 94.1 | **87.1** | **77.7** | **68.0** | 59.1 | **76.8** | 70.8 | 62.2 |

**Table 2:** Comparison of generalization ability across different network architecture with $(\varepsilon, \delta) = (10, 10^{-5})$. Our generated set is optimized with *ConvNet*, while the downstream classifiers are of different architecture. The classifiers are trained on the full synthetic set for baseline methods.

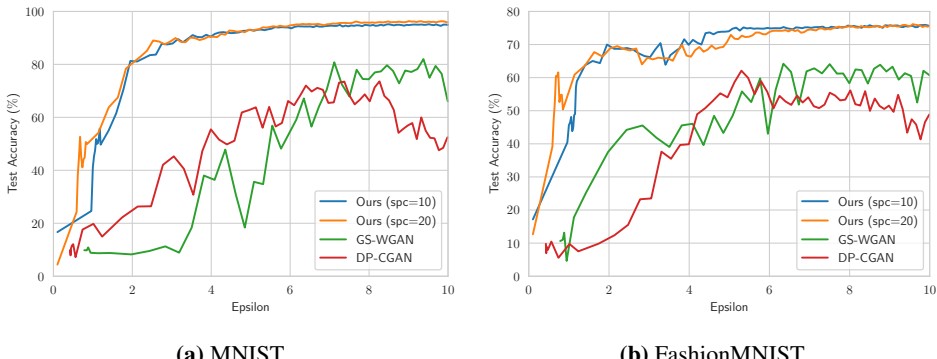

**(a)** MNIST  **(b)** FashionMNIST

**Figure 2:** Comparison of the convergence rate to existing private generative models with iteratively accumulated privacy cost. X-axis: privacy cost $\varepsilon$, Y-axis: utility (i.e., test accuracy (%)) for training downstream ConvNet classifiers.

to compress the useful information into a small set of samples and naturally save the memory and computation consumption for downstream analysis tasks.

**Generalization ability across different architectures.** One natural concern of our formulation could be the generalization ability to unseen situations. While we exploit discriminative information to guide the training, we (in principle) inevitably trade the generality off against task-specific utility, leaving no performance guarantees for new models. Interestingly, as shown in Table 2, we find that our generated set still provides better utility than all baseline methods in most cases, even though the models for evaluation have a completely different architecture from the one we used for training. The only case where our generated set does not work well is for training MLPs. We conjecture that it is due to the difference in the network properties that result in distinct gradient signals: for example, layers in MLPs are densely connected while being sparsely connected in ConvNets, and Convolutional layers are translation equivalent while FC layers in MLPs are not. Moreover, we argue that this may not be a bug, but a feature. Note that the reference results on real data also indicate that the MLP is inferior to other architectures while models with ConvNet, VGG, or ResNet architecture perform well in most cases. In this regard, results on our generated set generally align well with the result on real data, which suggests the possibility of conducting model selection with our private generated set.

**Convergence rate.** For most private (gradient-based) iterative methods, the privacy cost accumulates in each training iteration, and thus faster convergence is highly preferable. We show in Figure 2 the training curves where the y-axis denotes the utility and the x-axis corresponds to the privacy. We observe that our method generally has a much faster convergence rate than existing methods that need to accumulate the privacy cost for each iteration. In particular, our method already achieves a decent level of utility with $\varepsilon \leq 2$ which is much lower than the privacy budget used in most previous works (normally $\varepsilon = 10$).

## 5.2 Application: Private Continual Learning

The utility guarantee of our formulation requires that the network architecture is known to the data provider/generator. Fortunately, it is not a rare case in practice. In particular, our method is naturally

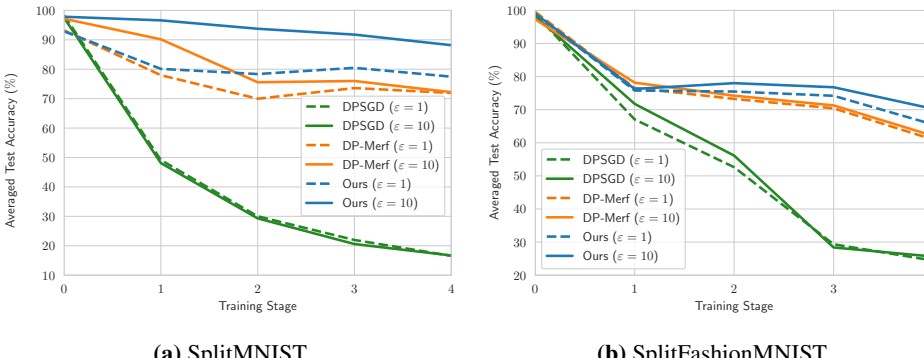

**(a)** SplitMNIST         **(b)** SplitFashionMNIST

**Figure 3:** Comparison for private training in the continual learning setting with $\delta = 10^{-5}$ and different $\varepsilon$. X-axis: training stage, Y-axis: averaged test accuracy (over all the stages till the current one). We use a ConvNet classifier in this case and set spc $= 10$ for our method and spc $= 6000$ for DP-Merf as default.

applicable to cases where *(i)* there are multiple parties involved in training a model and they agree on one common training protocol (i.e., the network architecture is known to all participants); *(ii)* each party has its own data whose privacy need to be protected (i.e., the training need to be DP); *(iii)* data on each party exhibit distinct property and is all informative for the final task (i.e., a synthetic set of representative samples that capture such properties would greatly aid the final task).

One example is continual learning [18, 29] where the training of the classification network is split into stages. Here we consider a setting adjusted to the DP training: to protect the privacy of its data, each party is responsible for a different training stage where it performs DP training of the model on its data, and subsequently delivers the DP model to the party responsible for the next training stage. Note that the raw data would not be transferred as otherwise the privacy would be leaked.

We conduct DP training on the SplitMNIST and SplitFashionMNIST datasets where the data is partitioned into 5 parts (based on the class labels, which corresponds to the class-incremental [27] setup) and we assume each part is held by one party and can not be accessed by others for privacy sake (See supplementary material for more details). We show in Figure 3 (green curves) the baseline results of DP training of model under the private class-incremental setting (i.e., each party finetune the model is obtained from the previous stage on its own data using DP-SGD). Apparently, this naive training scheme leads to catastrophic forgetting of information learned in the early stages. Even worse is that the common strategy to cope with this issue requires transferring a small set of real data to other parties such that it can be replayed in the later training stage [27, 3, 25], which is not directly applicable to the private setting as transferring the data breaks privacy. In contrast, private generation methods can be seamlessly applied to this case, where a set of DP synthetic samples is transferred to enable the final model to learn the characteristics of each partition of data. In particular, our formulation is better suitable for this setting than other generation methods as the network architecture is known to all participants and samples can be tailored to the specific network via our formulation. This is verified in Figure 3, where our synthetic samples are generally more informative for training the classifier when compared to DP-Merf – the overall best existing works in terms of the downstream utility. Moreover, as our formulation condenses the information into a small set of samples by construction, we also enjoy the advantages when considering the computation, storage, and communication cost.

## 6   Discussion

In this section, we present several key factors that distinguish our approach from existing ones and discuss possible concerns regarding our private set generation formulation.

**Trade-off between Visual Quality and Task Utility.** Our formulation is designed for optimizing the utility of downstream analysis tasks instead of the visual appearance as done in previous works, thereby leaving no performance guarantee for the visual quality of the synthetic samples. Moreover, the optimization of the private synthetic set is unconstrained and unregulated over the whole data space, with the gradient signal as the only guidance. As the data to gradient mapping is generally

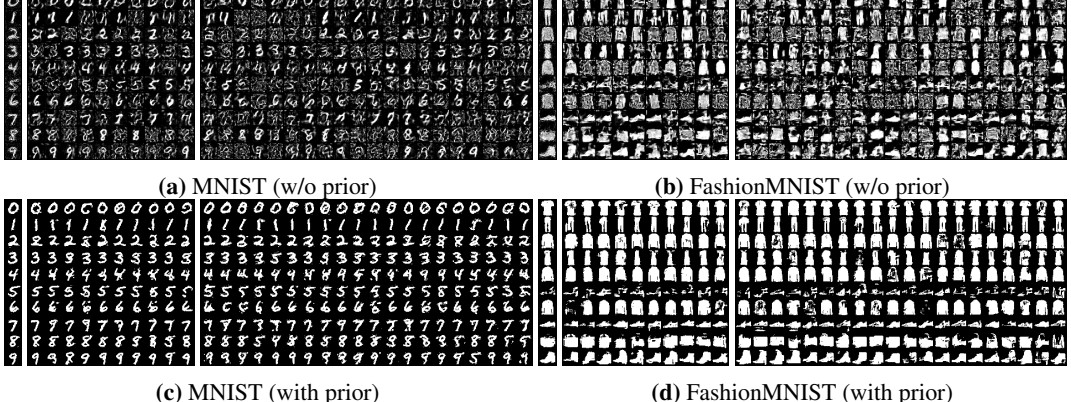

**(a)** MNIST (w/o prior)        **(b)** FashionMNIST (w/o prior)

**(c)** MNIST (with prior)        **(d)** FashionMNIST (with prior)

**Figure 4:** Our synthetic samples under $(\varepsilon, \delta) = (10, 10^{-5})$ for MNIST and FashionMNIST datasets with or without (w/o) incorporating a DCGAN generator network as image prior.

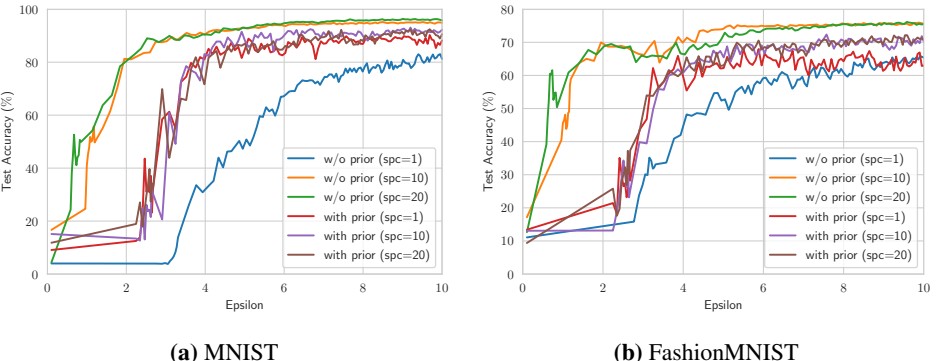

**(a)** MNIST        **(b)** FashionMNIST

**Figure 5:** Comparison of the convergence rate when training with or without (w/o) the image prior from DCGAN. X-axis: privacy cost $\varepsilon$, Y-axis: test accuracy (%) for training downstream ConvNet classifiers.

non-injective (i.e., different data can results in the same gradient), searching for the correct data given the gradient is an indefinite problem, which inevitably leads to outcomes that are out of the data manifold in practice. This can be seen in the first row of Figure 4 where we plot our private synthetic samples trained under the default setting.

Recall that one key difference between our formulation and existing works is that: we directly optimize for a set of samples instead of the deep generative models. We then take a further step and investigate whether this difference is the key factor that determines the samples' visual quality. To do this, we employ a DCGAN [26] model from [6] as the generator backbone (denoted as $G$), let $x^{\mathcal{S}}$ to be outputs of $G$, and then optimize over the network parameter of $G$ using the gradient matching loss as in Equation 5. Mathematically, this transforms Equation 7 into:

$$\min_{\boldsymbol{\varphi}} \mathbb{E}_{\boldsymbol{\theta}_0 \sim P_{\boldsymbol{\theta}_0}} \sum_{t=0}^{T-1} [\mathcal{L}_{\mathrm{dis}}(g_{\boldsymbol{\theta}_t}^{\mathcal{S}}, \widetilde{g_{\boldsymbol{\theta}_t}^{\mathcal{D}}})] \quad \text{with} \quad \mathcal{S} = \{G(\boldsymbol{z}_i; \boldsymbol{\varphi}), y_i^{\mathcal{S}}\}_{i=1}^{M} \tag{8}$$

where $\boldsymbol{\varphi}$ is the parameter of $G$, $\boldsymbol{z}_i$ is random Gaussian noise (fixed during training). Basically, this formulation restricts the synthetic images to be within the output space of $G$, and the inductive bias introduced by the convolutional structure serves as deep image prior [34] to regularize the visual appearance of the synthetic images.

We show the synthetic samples in the second row of Figure 4, and compare the utilities with our original formulation in Figure 5. We observe that the prior from the deep generative model is indeed important for the visual quality. However, interestingly, better visual quality does not mean better utility. Specifically, optimizing over the parameter of generator $G$ exhibits a slower convergence than directly optimizing the samples, while the final performance is also inferior (See quantitative results in Table 3). This gives several important indications that help inform future research in this field: *(i)* the goal of achieving better downstream utility may be incompatible with the goal of achieving

better sample visual quality, while dedicated efforts towards different goals are necessary; *(ii)* deep generative models may not be the best option for the task of private data generation as they result in suboptimal utility (mainly due to its slow convergence), which questions the current default way of thinking in this field.

**Scalability & Transparency.** We discuss the possible issues when scaling to more complicated datasets which: *(i)* contains a large number of label classes; *(ii)* are diverse and require a large number of samples to capture the statistical characteristics of the data distribution. For *(i)*, the complexity of our (and all the other) approaches will definitely increase as the number of label classes increases. When considering the number of variables that need to be optimized, the complexity increases linearly for our case, while for all methods (that optimize over the network parameters) the increase is no less than ours. While the application of DP deep learning (of discriminative models) to datasets with >10 label classes is rare, we anticipate that dealing with a much larger number of label classes is too ambitious for DP generative modeling for now. For *(ii)*, we conduct the experiment when varying the number of samples per class and present the results in 4, where we indeed observe the training difficulty when the number of samples increases. We conjecture that it is mainly because the gradient signals for updating the synthetic samples get sparser when the number increases, which results in a lower convergence rate and thus worse results especially when the allowed privacy budget is low. However, it is arguable whether this is a shortage as smaller amounts of samples allow more savings in the storage and computation consumption while providing greater transparency of downstream analysis.

| | MNIST | | | FashionMNIST | | |
|---|---|---|---|---|---|---|
| | 1 | 10 | 20 | 1 | 10 | 20 |
| w/o prior | 81.4 | **94.9** | **95.6** | 66.7 | **75.6** | **77.7** |
| with prior | **88.2** | 92.2 | 90.6 | 63.0 | 70.2 | 70.7 |

**Table 3:** Test accuracy (%) on real data of downstream ConvNet classifier with or without (w/o) adopting image prior from DCGAN under $(\varepsilon, \delta) = (10, 10^{-5})$.

| MNIST | | | | FashionMNIST | | | |
|---|---|---|---|---|---|---|---|
| 1 | 10 | 20 | 50 | 1 | 10 | 20 | 50 |
| 81.4 | 94.9 | 95.6 | 94.0 | 66.7 | 75.6 | 77.7 | 71.3 |

**Table 4:** Test accuracy (%) on real data of downstream ConvNet classifier when varying the numbers of samples per class (spc) under $(\varepsilon, \delta) = (10, 10^{-5})$.

**Generality and Expressiveness.** Our formulation focus on the task of training downstream neural networks, and thus have no guarantees for other (and more general) purpose. In contrast, deep generative models are designed for capturing the complete data distribution and, once perfectly trained, can be applied to more general cases. While our formulation seems to be inferior in this regard, we argue that this should not be a major shortcoming that outweighs the advantages: First of all, while deep generative models in principle have much greater expressiveness than a small set of samples, such upper bound is hard, if not impossible, to be achieved in the privacy learning setting. Instead, compromising the upper bound for a more achievable target is worthy and shows great improvement over existing works as demonstrated in section 5.1. Moreover, our formulation generalizes seamlessly to any gradient-based learning methods that a downstream analyst may adopt. While such methods already cover the most part of the possible analysis algorithms that could be adopted for high-dimensional data, we believe that our approach does exhibit a good level of practical applicability.

## 7 Conclusion

We introduce a novel view of private high-dimensional data generation: instead of attempting to train deep generative models in a DP manner, we directly optimize a set of samples under the supervision of discriminative information for downstream utility. We present a simple yet effective method that allows synthesizing a small set of samples that are representative of the original data distribution and informative for training downstream neural networks. We demonstrate via extensive experiments that our formulation leads to great improvement over state-of-the-art approaches in terms of the task utility, without losing the generality for performing analysis tasks in practice. Moreover, our results question the current default way of thinking and provide insights for further pushing the frontier in the field of private data generation. Our code has been open-sourced to facilitate research in the related field.

## Broader Impact

The widespread availability of rich data has fueled the growth of machine learning applications in numerous domains. However, in real-world application scenarios, data sharing is always prohibited due to the private nature of data and the corresponding stringent regulations, which greatly hinders technological progress. Our work contributes to making the latest advances in privacy-preserving data generation. In particular, our method improves the data utility compared to the state-of-the-art privacy-preserving data generation methods. In particular, we show the success on high dimensional data, which will be key to bringing those methods to a broader range of applications. Consequently, we expect broad adaptations of our technique and hence positive societal impacts. We are not aware of any extra negative societal impacts beyond generic risks of ML technology in general.

## Acknowledgments

This work is partially funded by the Helmholtz Association within the projects "Trustworthy Federated Data Analytics (TFDA)" (ZT-I-OO1 4), and "Protecting Genetic Data with Synthetic Cohorts from Deep Generative Models (PRO-GENE-GEN)" (ZT-I-PF-5-23). Additionally, this work is supported in part by ELSA - European Lighthouse on Secure and Safe AI funded by the European Union under grant agreement No. 101070617. Views and opinions expressed are however those of the authors only and do not necessarily reflect those of the European Union or European Commission. Neither the European Union nor the European Commission can be held responsible for them. We also acknowledge Max Planck Institute for Informatics and "Helmholtz AI computing resources" (HAICORE) for providing computing resources.

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
