# Supplementary Materials for "Private Set Generation with Discriminative Information"

These supplementary materials include the privacy analysis (§A), the details of the adopted algorithms (§B), and the details of experiment setup (§C), and additional results and discussions (§D). The source code is available at https://github.com/DingfanChen/Private-Set.

## A  Privacy Analysis

Our privacy computation is based on the notion of Rényi-DP, which we recall as follows.

**Definition A.1.** (Rényi Differential Privacy (RDP) [8]). A randomized mechanism $\mathcal{M}$ is $(\alpha, \varepsilon)$-RDP with order $\alpha$, if

$$D_\alpha(\mathcal{M}(\mathcal{D})\|\mathcal{M}(\mathcal{D}')) = \frac{1}{\alpha - 1} \log \mathbb{E}_{x \sim \mathcal{M}(\mathcal{D})} \left[ \left( \frac{Pr[\mathcal{M}(\mathcal{D}) = x]}{Pr[\mathcal{M}(\mathcal{D}') = x]} \right)^{\alpha - 1} \right] \le \varepsilon \qquad (1)$$

holds for any adjacent datasets $\mathcal{D}$ and $\mathcal{D}'$, where $D_\alpha(P\|Q) = \frac{1}{\alpha-1} \log \mathbb{E}_{x \sim Q}[(P(x)/Q(x))^\alpha]$ is the Rényi divergence of order $\alpha > 1$ between the distributions $P$ and $Q$.

To compute the privacy cost of our approach, we numerically compute $D_\alpha(\mathcal{M}(\mathcal{D})\|\mathcal{M}(\mathcal{D}'))$ in Definition A.1 for a range of orders $\alpha$ [9, 14] in each training step that requires access to the real gradient $g_{\boldsymbol{\theta}}^{\mathcal{D}}$. To obtain the overall accumulated privacy cost over multiple training iterations, we use the composition properties of RDP summarized by the following theorem.

**Theorem A.1.** (Adaptive Composition of RDP [9]). Let $f : \mathcal{D} \to \mathcal{R}_1$ be $(\alpha, \varepsilon_1)$-RDP and $g : \mathcal{R}_1 \times \mathcal{D} \to \mathcal{R}_2$ be $(\alpha, \varepsilon_2)$-RDP, then the mechanism defined as $(X, Y)$, where $X \sim f(\mathcal{D})$ and $Y \sim g(X, \mathcal{D})$, satisfies $(\alpha, \varepsilon_1 + \varepsilon_2)$-RDP

In total, our private set generation (PSG) approach (shown in Algorithm 1 of the main paper) and the generator prior variant (shown in Algorithm 2) can be regarded as a composition over $RTK$ (i.e., the number of iterations where the real gradient is used) homogenous subsampled Gaussian mechanisms (with the subsampling ratio $= B/N$) in terms of the privacy cost.

Lastly, we use the following theorem to convert $(\alpha, \varepsilon)$-RDP to $(\varepsilon, \delta)$-DP.

**Theorem A.2.** (From RDP to $(\varepsilon, \delta)$-DP [8]). If $\mathcal{M}$ is a $(\alpha, \varepsilon)$-RDP mechanism, then $\mathcal{M}$ is also $(\varepsilon + \frac{\log 1/\delta}{\alpha - 1}, \delta)$-DP for any $0 < \delta < 1$.

## B  Algorithms

**Objective** . The distance $\mathcal{L}_{\text{dis}}$ (in Equation 5 of the main paper) between the real and synthetic gradients is defined to be the sum of cosine distance at each layer [20, 18]. Let $\boldsymbol{\theta}^l$ denote the weight at the $l$-th layer, the distance can be formularized as follows,

$$\mathcal{L}_{\text{dis}}(\nabla_{\boldsymbol{\theta}}\mathcal{L}(\mathcal{S}, \boldsymbol{\theta}_t), \nabla_{\boldsymbol{\theta}}\mathcal{L}(\mathcal{D}, \boldsymbol{\theta}_t)) = \sum_{l=1}^{L} d(\nabla_{\boldsymbol{\theta}^l}\mathcal{L}(\mathcal{S}, \boldsymbol{\theta}_t), \nabla_{\boldsymbol{\theta}^l}\mathcal{L}(\mathcal{D}, \boldsymbol{\theta}_t))$$

where $d$ denotes the cosine distance between the gradients at each layer:

$$d(\boldsymbol{A}, \boldsymbol{B}) = \sum_{i=1}^{out} \left( 1 - \frac{\boldsymbol{A}_{i\cdot} \cdot \boldsymbol{B}_{i\cdot}}{\|\boldsymbol{A}_{i\cdot}\|\|\boldsymbol{B}_{i\cdot}\|} \right)$$

$\boldsymbol{A}_{i\cdot}$ and $\boldsymbol{B}_{i\cdot}$ are the flattened gradient vectors to each output node $i$. For FC layers, $\boldsymbol{\theta}^l$ is a 2D tensor with dimension $out \times in$ and the flattened gradient vector has dimension $in$, while for Conv layer, $\boldsymbol{\theta}^l$ is a 4D tensor with dimensionality $out \times in \times h \times w$ and the flattened vector has dimension $in \times h \times w$. $out, in, h, w$ corresponds to the number of output and input channels, kernel height, and width, respectively.

**Generator Prior** . We present the pseudocode of the generator prior experiments (Section 6 of the main paper) in Algorithm 2, which is supplementary to Figure 4,5 and Equation 8 of the main paper.

---

**Algorithm 2:** Private Set Generation with Generator Prior

---

**Input:** Dataset $\mathcal{D} = \{(\boldsymbol{x}_i, y_i)\}_{i=1}^N$, learning rate for update network parameters $\tau_{\boldsymbol{\theta}}$ and $\tau_{\boldsymbol{\varphi}}$, batch size $B$, DP noise scale $\sigma$, gradient clipping bound $C$, number of runs $R$, outer iterations $T$, inner iterations $J$, batches $K$, number of classes $L$, number of samples per class (spc), desired privacy cost $\varepsilon$ given a pre-defined $\delta$

**Output:** Synthetic set $\mathcal{S}$

Compute the DP noise scale $\sigma$ numerically so that the privacy cost equals to $\varepsilon$ after the training;
Initialize model parameter $\boldsymbol{\varphi}$ of the conditional generator $G$;

**for** $c$ **in** $\{1, ..., L\}$ **do**
  **for** *sample_index* **in** spc **do**
    $y_i^{\mathcal{S}} = c$ ;
    Sample $\boldsymbol{z}_i \sim \mathcal{N}(0, I)$ ($\boldsymbol{z}_i$ is fixed for each corresponding synthetic sample during the training) ;
    $\boldsymbol{x}^{\mathcal{S}} = G(\boldsymbol{z}_i, y_i^{\mathcal{S}}; \boldsymbol{\varphi})$;
    Insert $(\boldsymbol{x}_i^{\mathcal{S}}, y_i^{\mathcal{S}})$ into $\mathcal{S}$;
  **end**
**end**

**for** *run* **in** $\{1, ..., R\}$ **do**
  Initialize model parameter $\boldsymbol{\theta}_0 \sim P_{\boldsymbol{\theta}_0}$;
  **for** *outer_iter* **in** $\{1, ..., T\}$ **do**
    $\boldsymbol{\theta}_{t+1} = \boldsymbol{\theta}_t$
    **for** *batch_index* **in** $\{1, ..., K\}$ **do**
      Uniformly sample random batch $\{(\boldsymbol{x}_i, y_i)\}_{i=1}^B$ from $\mathcal{D}$;
      **for** *each* $(\boldsymbol{x}_i, y_i)$ **do**
        `// Compute per-example gradients on real data`
            $g_{\boldsymbol{\theta}_t}^{\mathcal{D}}(\boldsymbol{x}_i) = \ell(F(\boldsymbol{x}_i; \boldsymbol{\theta}_t), y_i)$
        `// Clip gradients`
            $\widetilde{g_{\boldsymbol{\theta}_t}^{\mathcal{D}}}(\boldsymbol{x}_i) = g_{\boldsymbol{\theta}_t}^{\mathcal{D}}(\boldsymbol{x}_i) \cdot \min(1, C/\|g_{\boldsymbol{\theta}_t}^{\mathcal{D}}(\boldsymbol{x}_i)\|_2)$
      **end**
      `// Add noise to average gradient with Gaussian mechanism`
        $\widetilde{g_{\boldsymbol{\theta}_t}^{\mathcal{D}}} = \frac{1}{B} \sum_{i=1}^B (\widetilde{g_{\boldsymbol{\theta}_t}^{\mathcal{D}}}(\boldsymbol{x}_i) + \mathcal{N}(0, \sigma^2 C^2 I))$
      `// Compute parameter gradients on synthetic data and update` $G$
        $g_{\boldsymbol{\theta}_t}^{\mathcal{S}} = \nabla_{\boldsymbol{\theta}} \mathcal{L}(\mathcal{S}, \boldsymbol{\theta}_t)) = \frac{1}{M} \sum_{i=1}^M \ell(F(\boldsymbol{x}_i^{\mathcal{S}}; \boldsymbol{\theta}_t), y_i^{\mathcal{S}})$ where $\boldsymbol{x}_i^{\mathcal{S}} = G(\boldsymbol{z}_i, y_i^{\mathcal{S}}; \boldsymbol{\varphi})$
        $\boldsymbol{\varphi} = \boldsymbol{\varphi} - \tau_{\boldsymbol{\varphi}} \cdot \nabla_{\boldsymbol{\varphi}} \mathcal{L}_{\mathrm{dis}}(g_{\boldsymbol{\theta}_t}^{\mathcal{S}}, \widetilde{g_{\boldsymbol{\theta}_t}^{\mathcal{D}}})$
    **end**
  **end**
  **for** *inner_iter* **in** $\{1, ..., J\}$ **do**
    `// Update network parameter using` $\mathcal{S}$
      $\mathcal{S} = \{G(\boldsymbol{z}_i, y_i^{\mathcal{S}}; \boldsymbol{\varphi}), y_i^{\mathcal{S}}\}_{i=1}^M$
      $\boldsymbol{\theta}_t = \boldsymbol{\theta}_t - \tau_{\boldsymbol{\theta}} \cdot \nabla_{\boldsymbol{\theta}} \mathcal{L}(\mathcal{S}, \boldsymbol{\theta}_t)$
  **end**
**end**
**return** Synthetic set $\mathcal{S}$

---

The only difference to the original PSG formulation is that the samples are restricted to be the output of a generator network and the updates are conducted on the generator network parameters (See Figure 1 for the illustration and see Figure 1 in the main paper for a comparison). Note that we fix the random latent code $z_i$ during the whole training process to guarantee that there is no other randomness/degree of freedom except that introduced by the generator network itself. While it is possible to allow random sampling of the latent code and generate change-

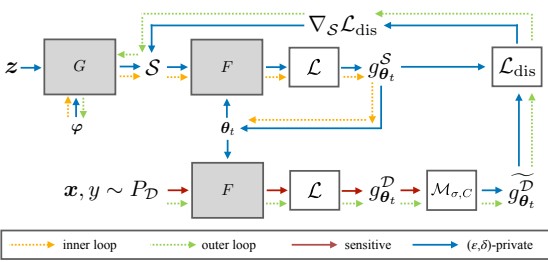

**Figure 1:** Training pipeline (with prior).

able $\mathcal{S}$ to mimic the training of generative models (i.e., train a generative network using the gradient matching loss), we observe that the training easily fails in the early stage. We argue that this also indicates that training a generative network is a harder task than training a set of samples directly, which explains the better convergence behavior and superior final performance of our formulation in comparison to existing works (which build on top of deep generative networks).

## C Experiment Setup

### C.1 Datasets

**MNIST** [6] dataset contains $28 \times 28$ grayscale images of digit numbers. The dataset comprises 60K training images and 10K testing images in total. The task is to classify the image into one of the 10 classes based on the digit number it contains.

**Fashion-MNIST** [15] dataset consists of $28 \times 28$ grayscale images fashion products of 10 categories. The total dataset size is 60K for the training set and 10K for the testing set, respectively. The task is to classify the fashion product given in the images.

### C.2 Required Resources and Computational Complexity

All our models and methods are implemented in PyTorch. Our experiments are conducted with Nvidia Tesla V100 and Quadro RTX8000 GPUs and a common configuration with 16GB GPU memory is sufficient for conducting all our experiments.

In comparison to normal non-private training, the major part of the additional memory and computation cost is introduced by the DP-SGD [1] step (for the per-sample gradient computation) that sanitizes the parameter gradient on real data, while the other steps (including the update on $\mathcal{S}$, and the updates of $F(\cdot; \boldsymbol{\theta})$ on $\mathcal{S}$ are equivalent to multiple calls of the normal non-private forward and backward passes (whose costs have lower magnitude than the DP-SGD step). Moreover, our formulation requires much less computational and memory consumption than previous works that require training multiple instances of the generative modules [3, 7, 12].

### C.3 Hyperparameters

**Training.** We set the default value of hyperparameters as follows: batch size = 256 for both computing the parameter gradients in the outer iterations and for update the classifier $F$ in the inner iterations, gradient clipping bound $C = 0.1$, $R = 1000$ for $\varepsilon = 10$ (and $R = 200$ for $\varepsilon = 1$), $K = 10$. The number of inner $J$ and outer $T$ iterations are dependent on the number of samples per class ($spc$), as more samples generally requires more iterations till convergence: $(T, J)$ is set to be $(1, 1)$, $(10, 50)$, $(20, 25)$ and $(50, 10)$ for $spc = 1, 10, 20, 50$, respectively. The DP noise scale $\sigma$ is calculated numerically[1] [1, 9] so that the privacy cost equals to $\varepsilon$ after the training (with $RTK$ steps in total that consume privacy budget), given that $\delta = 10^{-5}$. The learning rate is set to be $\tau_{\boldsymbol{\theta}} = 0.01$ (and $\tau_{\boldsymbol{\varphi}} = 0.01$ for training with generator prior) and $\tau_{\mathcal{S}} = 0.1$ for updating the network parameters and samples, respectively. We use SGD optimizer for the classifier $F$, and samples $\mathcal{S}$

---

[1] Based on Google's TensorFlow privacy under version $\leq 0.8.0$: https://github.com/tensorflow/privacy/blob/master/tensorflow_privacy/privacy/analysis/rdp_accountant.py

(with momentum= $0.5$), while we use Adam optimizer for the generator $G$ if trained with prior. For the training process, no data augmentation is adopted. Our implementation of the DP-SGD step and the uniform data sampling operation is based on the Opacus [16] [2] package.

**Evaluation.** We set the epoch to be $40$ and $300$ when training the downstream classification models on the synthetic data with "full" size ($spc = 6000$) and small size ($spc \in \{1, 10, 20, 50\}$), respectively, to guarantee the convergence of downstream model training and maintain the evaluation efficiency. We set the learning rate to be $0.01$ at the beginning and decrease it (by multiplying with $0.1$) when half of the total epoch is achieved. We use SGD optimizer with momentum= $0.9$, weight decay= $5 \cdot 10^{-4}$ and set batch size = $256$ for training the classifier. Random cropping and re-scaling are adopted as data augmentation when training the classification model.

## C.4  Baseline Methods

We present more details about the implementation of the baseline methods. In particular, we provide the default value of the privacy hyperparameters below.

**DP-Merf  [4]** [3] For $\varepsilon = 1$ we use as the default hyperparameters setting provided in the official implementation: DP noise scale $\sigma = 5.0$, training epoch = $5$, while for $\varepsilon = 10$, the DP noise scale is $\sigma = 0.568$.

**DP-CGAN [11]** [4] We set the default hyper-parameters as follows: gradient clipping bound $C = 1.1$, noise scale $\sigma = 2.1$, batch size= $600$ and total number of training iterations= $30K$. We exclude this model from evaluation at $\varepsilon = 1$ as the required noise scale is too large for the training to make progress, which is consistent with the results in literature [4, 3].

**GS-WGAN [3]**  [5] We adopt the default configuration provided by the official implementation ($\varepsilon = 10$): the subsampling rate = $1/1000$, DP noise scale $\sigma = 1.07$, batch size = $32$. Following [3], we pretrain (warm-start) the model for $2K$ iterations, and subsequently train for 20K iterations. Similar to the case for DP-CGAN, we exclude this model from evaluation at $\varepsilon = 1$ as the required noise scale is too large for the training to be stable.

For **G-PATE** [7], **DataLens** [12] and **DP-Sinkhorn** [2], we present the same results as reported in the original papers (in Table 1 of the main paper) as reference, as they are either not directly comparable to ours or not open-sourced.

## C.5  Private Continual Learning

**Setting.** The experiments presented in Section 5.2 of the main paper correspond to the class-incremental learning setting [10] where the data partition at each stage contains data from disjoint subsets of label classes. And the task protocol is sequentially learning to classify a given sample into all the classes seen so far. For our experiments on SplitMNIST and SplitFashionMNIST benchmarks [17], the datasets are split into 5 partitions each containing samples of 2 label classes. The evaluation task is thus binary classification for the first stage, while two more classes are included after each following stage.

While a clear definition of the private continual learning setting is, to the best of our knowledge, missing in the literature, we introduce a basic case where privacy can be strictly protected during the whole training process. In brief, we need to guarantee that all the information that is delivered to another party/stage should be privacy-preserving.

Hence, for the **DP-SGD** [1] baseline, the classification model is initialized to be a 10-class classifier, and is updated (fine-tuned) via DP-SGD at each training stage on each data partition. During the whole process, the model is transferred between different parties while privacy is guaranteed by DP-SGD training.

---

2    https://opacus.ai/
3    https://github.com/frhrdr/dp-merf
4    https://github.com/reihaneh-torkzadehmahani/DP-CGAN
5    https://github.com/DingfanChen/GS-WGAN

And for the private generation methods, i.e., **DP-Merf** and **Ours**, we use a fixed privacy budget to train a private generative model or a private synthetic set for each partition/stage. Subsequently, such a generative model or synthetic set is transferred between parties for conducting different training stages. For evaluation, a $n$-class classifier is initialized and then trained on the transferred private synthetic samples for each stage, where $n$ is the total number of label classes seen so far. In our experiments, both methods only exploit information from the local partition for the generation, i.e., our private set is optimized on a freshly initialized classification network at each stage and for DP-Merf the mean embedding is taken over the local partition. While our formulation can be adjusted to (and may be further improved by) more advanced training strategies designed for continual learning to eliminate forgetting, many of such strategies are not directly compatible with private training as they require access to old data. We believe that our introduced private continual learning setting is of independent interest and leave an in-depth investigation of this topic as future work.

**Hyperparameters.** We use the default values for the hyperparameters as shown in Section C.3 and C.4, except that the training epoch is set to be 10 for **DP-SGD** and the runs $R = 200$ for **Ours**, to balance the convergence, forgetting effect, and evaluation efficiency. Moreover, the DP noise scale is calibrated to each *partition* of the data.

# D    Additional Results and Discussions

## D.1    Dataset Distillation Basis

In this paper, we propose to use the gradient matching technique [20, 18] (among existing dataset distillation approaches) as a basis for private set generation. In the following, we briefly discuss other popular dataset condensation approaches that achieve competitive performance for non-private tasks but appear less suitable for private learning. For example, [13] requires solving a nested optimization problem, which makes it hard to quantify the individual's effect (i.e., the sensitivity) and thus difficult to impose DP into the training. In addition, [19] relies on "per-class" feature aggregation as the only source of supervision to guide the synthetic data towards representing its target label class. However, this "per-class" operation contradicts label privacy and the requirement of uniform sampling for the privacy cost computation. In contrast, our formulation adopts uniform sampling (which is compatible with DP) and exploits the (inherently class-dependent) gradient signals to generate representative samples.

## D.2    Computation Time

Under the default setting (See Section C.2 and C.3), it takes around 4.5 hours and 11 hours to train the synthetic data for the case of $spc = 10$ and $spc = 20$, respectively. To the best of our knowledge, our method is more efficient than existing works that require pre-training of (multiple) models [3, 7], but requires more running time than methods that use static pre-computed features [4]. Moreover, we see a tendency that the distilled dataset requires less time on downstream tasks compared to samples from generative models due to the smaller (distilled) sample size.

## D.3    Evaluation on Colored Images

In this section, we provide additional evaluation results on colored image benchmark dataset. On CIFAR-10 [5] dataset, We use the same default setting as described in Section C.3 and adjust the network architectures to the input dimension $(32 \times 32 \times 3)$. We summarize in Figure 1 the quantitative results of downstream utility when varying the number of samples per class ($spc \in \{1, 10, 20\}$)

|  | 1 | 10 | 20 |
|---|---|---|---|
| non-private | 30.0 | 48.6 | 52.6 |
| $\varepsilon = 10$ | 28.9 | 40.3 | 42.6 |

**Table 1:** Test accuracy (%) on real data of downstream ConvNet classifier on CIFAR-10.

and show as reference the results when training non-privately (We show here the results when applying uniform sampling of the data instead of the original per-class sampling approach [20, 18] also for the non-private baseline for controlled comparison). Additionally, we show in Figure 2 the synthetic images when training under DP ($\varepsilon = 10$), and in 3 the results when training non-privately. We observe that while the synthetic samples look noisy and non-informative, they do provide useful features for downstream classifiers, leading to a decent level of performance. Note that colored

images are generally challenging for private learning. In fact, this makes our work the first one that is able to report non-trivial performance on this dataset.

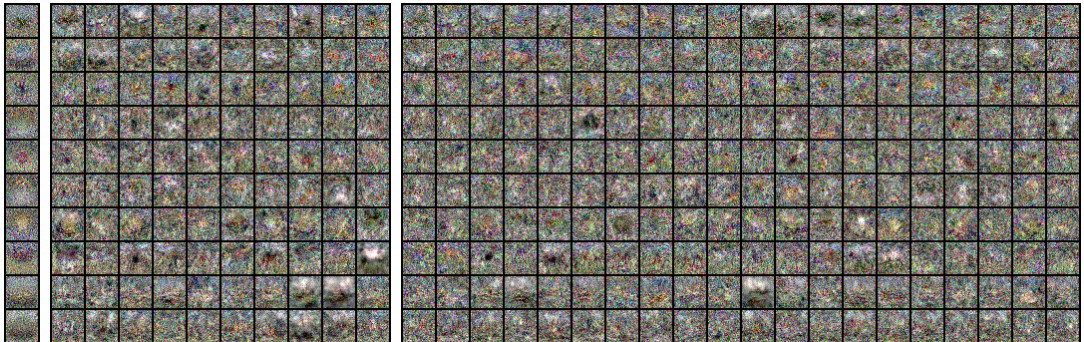

**Figure 2:** CIFAR-10 ($\varepsilon = 10$)

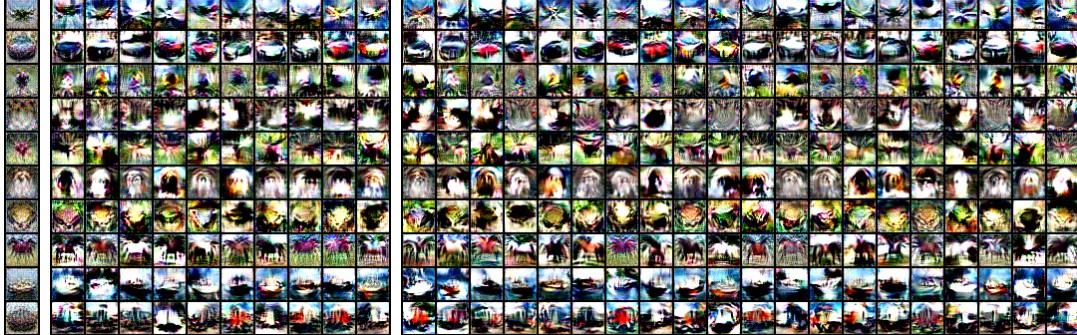

**Figure 3:** CIFAR-10 (non-private)