# OpenReview forum: "Private Set Generation with Discriminative Information"
_NeurIPS.cc/2022/Conference — NeurIPS 2022 Accept_

### Official Review · Reviewer_b3jX · 2022-06-15

**Rating:** 7
**Confidence:** 3
**Soundness:** 3 good
**Presentation:** 3 good
**Contribution:** 3 good

**Summary:**

This paper proposes a new utility-optimized approach of privatizing high-dimensional data generation which optimizes the synthetic samples instead of the deep generative models. Experiments on MNIST and FashionMNIST are conducted to demonstrate the effectiveness.

**Questions:**

I only have two minor questions.

1) In algorithm 1, what equation is used to determine the value of $\sigma$ in order to preserve $(\epsilon, \delta)$-DP? I think this may need to be explicitly discussed.

2) What are the numerical values in terms of the computational cost? I checked the supplementary but found this is missing. In particular, I am interested in the algorithm running time for different values of spc, say 10, 20, 6000 for MNIST.

**Limitations:**

The authors adequately discussed the low visual quality issue of the proposed approach. Other aspects such as scalability and generality are also mentioned.

**Strengths And Weaknesses:**

Originality. The problem that the paper aims to address is well motivated. The proposed approach seems new to me. Relevant literatures are well discussed.

Quality. The proposed algorithm overall makes sense to me and I didn't find any technical mistakes. However, some details of the algorithm and the experiments are missing (see my questions below).

Clarity. The paper is easy to follow and well organized.

Significance. The results overall look good, yet I am not sure whether they are enough. The authors only chose $\epsilon = 1$ and 10, corresponding to high and low privacy budget. Not sure whether what happens when the budget is moderate, say $\epsilon = 5$. Also, it would be better if the performance is benchmarked on colored image datasets (e.g., CIFAR-10).

---

> ### Author Response · Authors · 2022-08-02
> **Response to Reviewer b3jX**
>
> We thank all reviewers for providing valuable and constructive feedback. We are encouraged that all reviewers have given positive comments, finding our work **" presents an interesting novel perspective"** and **“could have a large impact”**, the paper is **"very well written"** and **"easy to follow and well organized"**, the experimental evaluation is **"extensive"** and **"improves the SOTA data utility by a large margin"**.
>
> We now address individual concerns of **Reviewer b3jX**. All minor points are incorporated in the revised manuscript, where the modified text is marked in **blue**.
>
> 1. **[The authors only chose $\epsilon=1$ and 10, corresponding to high and low privacy budgets. Not sure whether what happens when the budget is moderate, say $\epsilon=5$. ]**
> The behavior of our method across a wide range of privacy budgets (~0 to 10) can be seen in Figure 2 in the main paper, where we compare the privacy-utility trade-off of different methods. We can observe that our method already achieves promising performance (i.e., >90% test accuracy for MNIST, and >70% for FashionMNIST) under a moderate budget ($\epsilon \approx 5$).
> &nbsp;
> &nbsp;
>
> 2. **[Also, it would be better if the performance is benchmarked on colored image datasets (e.g., CIFAR-10).]**
> We provide additional experimental results on CIFAR-10 dataset in the revised supplementary material (Section D.3) and summarize the quantitative results below. Please note that (high-dimensional) colored images are generally hard for private learning. In fact, this makes our work the first work that is able to report non-trivial performance on this dataset.
> | spc= | 1 | 10 | 20 |
> |-------------|------|------|------|
> | non-private | 30.0 | 48.6 | 52.6 |
> | epsilon=10 | 28.9 | 40.3 | 42.6 |
>
>
> 3. **[In algorithm 1, what equation is used to determine the value of $\sigma$ in order to preserve $(\epsilon,\delta)$-DP?]**
> We provide the corresponding references (paper and code) in the revised version of the main paper (Algorithm 1) and supplementary material (Page 4 footnote 1).
> &nbsp;
> &nbsp;
>
> 4. **[What are the numerical values in terms of the computational cost? I checked the supplementary but found this is missing. In particular, I am interested in the algorithm running time for different values of spc, say 10, 20, 6000 for MNIST.]**
> We provide the running time evaluation in the revised supplementary material (Section D.2). Under the default setting, it takes around 4.5 hours and 11 hours to train the synthetic data for the case of spc=10 and spc=20, respectively. To the best of our knowledge, our method is more efficient than existing works that require pre-training of (multiple) models (e.g., GS-WGAN, G-PATE), but requires more running time than methods that use static pre-computed features (e.g., DP-Merf).

---

> > ### Comment · Reviewer_b3jX · 2022-08-03
> > **Thank the authors for revising the paper**
> >
> > Thank the authors for revising the paper. I am satisfied with the authors' comments as well as the latest changes, which further improve the quality of the paper. Hence I changed my rating from 6 to 7.

---

### Official Review · Reviewer_GR88 · 2022-07-07

**Rating:** 7
**Confidence:** 4
**Soundness:** 3 good
**Presentation:** 3 good
**Contribution:** 3 good

**Summary:**

This paper introduces a new approach to creating a synthetic data set as a stand-in for releasing some sensitive data. Instead of the standard technique of learning some kind of general purpose generative model that tries to capture the full data distribution, the proposed methods iteratively updates a set of data points (initialised randomly) directly to maximise the utility of the synthetic data for a given downstream classification task.

More precisely, the synthetic data is used to train the downstream classification network while in parallel the original data is used to train a network of the same architecture. In each step, the synthetic data is then updated to minimise the difference between the network parameter gradients between the two networks undergoing training. Privacy is ensured by perturbing the gradients of the network trained on the original data using the Gaussian mechanism before it is used in optimising the synthetic data.

The paper performs an extensive evaluation of the method and compares it to competing methods from recent literature on image classification tasks for MNIST and FashionMNIST using a convolutional network. The classifier trained on synthetic data generated using the proposed method shows superior classification accuracy compared to when trained on synthetic data generated using competing methods. Additionally, the proposed method allows to sample a much smaller data set than the original without a significant loss in downstream accuracy, which makes downstream training much easier.

The paper additionally explores using different architectures for the downstream task, which is found to have no significant effects on accuracy as long as the architecture is sufficiently similar (i.e., some kind of convolutional network).

Finally, the paper discusses that the generated synthetic data may fall out of the original data manifold due to unconstrained optimisation, and therefore does not "look like" the original data. The discussion includes a modification of the proposed method that ensures that the synthetic data is on the manifold and compares it to the unconstrained method.

**Questions:**

- I cannot comprehend what the sentence in lines 141-143 is trying to say, can you elaborate?

Minor notes/nitpicks:
- Def 3.1 : "There are two ways in which $\mathcal{D}$ and $\mathcal{D}'$ may differ in one training example: it could be entirely removed or replaced by a different one. This makes a (small) difference, so please specify your setting (or state explicitly that it does not matter in your case).
- Thm 3.1 : "Any function F"; technically the function $F$ is constrained to those that do not access the original data again.

**Limitations:**

The proposed method has a number of limitations that could hamper its adoption in practice. The paper discusses and performs experiments exploring the major ones of these but not always to a full degree:

- The method requires a downstream task to be known a-priori before generating the data. This may not always be the case, however it is probably likely for the data holder to find the "most likely" task their data will be used for (they have collected that data for a reason, after all). The authors give an example in Sec 5.2 of a use case for their method in practice.

- Generalisability: The synthetic data produced using this method has no guarantee to be useful for anything but the downstream task it was optimised for. The authors recognise this and perform additional experiments in Sec 5.1 where they test different downstream classification architectures. They find that utility is similar for different architectures of convolutional networks (with the synthetic data optimised using a CNN as well) but drops off for MLP networks; no other architectures are explored. The authors do not evaluate different downstream tasks that could be performed on the synthetic data, so it is remains somewhat unclear how the synthetic data generalises. However, the authors point out in the end of Sec. 6. that while other methods may have in theory better capacity to generalise, these fail to show that that materialises as a benefit in any particular task in practise.

- Out-of-distribution synthetic data: The synthetic data is optimised in an unconstrained domain and may fall out of the manifold that the original data lies on. The authors discuss and evaluate this and provide the interesting finding that there appears to be a trade-off between generating data that lies on the correct manifold versus data that maximises utility. There is no deeper inspection of why that is the case: It remains especially unclear whether this is merely an artifact of compressing the original data into a smaller synthetic set or an unavoidable trade-off. This also ties in with generalisability discussed above: A task other than the chosen downstream task may require data to be more similar to the original data (such as, e.g., image segmentation in this case).

Summarising: The method does have limitations and the paper discusses the major ones. While some questions remain and follow up work on these would be welcome, they may be deemed to lie outside the permissible length of a conference paper and the discussion present should be sufficient for the reader to be aware of the trade-offs and whether the method is suitable in their setting.

**Strengths And Weaknesses:**

Originality: This paper presents an interesting novel perspective on generating synthetic data that deviates from established practise for generating synthetic data under DP guarantees. However, since I am not closely familiar with the privacy-agnostic(/non-private) data distillation technique, it could be that these contain similar methods except for the privacy perturbation, which could make this work a somewhat iterative. I hope some of my fellow reviewers may be more familiar with that and can form a more well-founded opinion on this. However, while the method would be iterative in that case, I believe that the evaluation for the privacy setting would still be a sufficiently original contribution.

Significance: The proposed methods demonstrates clearly superior results for the chosen downstream tasks, which could have a large impact for sensitive data release in practical applications when the downstream task the data will be used in is clear (or can be deduced with high likelihood). While I do not know how likely that setting is, I do think that it is not one impossible to find oneself in, so designing a well-working method for such a case is a valuable contribution.

Soundness/Quality: The proposed methods applies standard privacy techniques in a straightforward way, so results on privacy from earlier work transfer more or less directly. The experiments appear to have no technical flaws, however the reported results do not convey any information on error margins.

Clarity: The paper is overall clear and easy to understand.

---

> ### Author Response · Authors · 2022-08-02
> **Response to Reviewer GR88**
>
> We thank all reviewers for providing valuable and constructive feedback. We are encouraged that all reviewers have given positive comments, finding our work **" presents an interesting novel perspective"** and **“could have a large impact”**, the paper is **"very well written"** and **"easy to follow and well organized"**, the experimental evaluation is **"extensive"** and **"improves the SOTA data utility by a large margin"**.
>
> We now address individual concerns of **Reviewer GR88**. All minor points are incorporated in the revised manuscript, where the modified text is marked in **blue**.
>
> 1. **[The authors do not evaluate different downstream tasks that could be performed on the synthetic data, so it remains somewhat unclear how the synthetic data generalises. ]**
> We agree that evaluating different downstream tasks would be an important direction for the community. However, classification is the only downstream task – yet a very important and prominent one - that has been studied in existing works of DP high-dimensional data generation. While it would be particularly interesting to incorporate different downstream tasks into the design of the training objective, it is in fact difficult given the current status of research and would require future exploration.
> &nbsp;
> &nbsp;
> 2. **[It remains especially unclear whether the trade-off (between generating data that lies on the correct manifold versus data that maximises utility) is merely an artifact of compressing the original data into a smaller synthetic set or an unavoidable trade-off. This also ties in with generalisability discussed above: A task other than the chosen downstream task may require data to be more similar to the original data (such as, e.g., image segmentation in this case).]**
> It is indeed a general question whether there exists a fundamental trade-off between generating data that lies on the data manifold versus data that maximizes utility. While we provide some insights via our study using an image prior, we have only scratched the surface here and more research is required to fully address this open question.
> &nbsp;
> &nbsp;
> 3. **[Clarification of lines 141-143, Def 3.1, and Thm 3.1.]**
> We thank the reviewer for pointing these out. Please check the revised submission, where we have rephrased and provided more information accordingly.

---

> > ### Comment · Reviewer_GR88 · 2022-08-09
> >
> > I thank the authors for considering my and the other reviewers' comments, updating the paper correspondingly and providing additional information.
> >
> > I will most likely keep my current recommendation and score.

---

### Official Review · Reviewer_qnNn · 2022-07-10

**Rating:** 6
**Confidence:** 4
**Soundness:** 3 good
**Presentation:** 4 excellent
**Contribution:** 2 fair

**Summary:**

This paper proposes to use recent advances for dataset condensation as an alternative for differentially private data generation. Specifically, they use DP-SGD to optimize a gradient matching objective that minimizes the distance between the gradients of real data on the target task and the gradients of synthetic data on the target task. The authors show this approach generates better data, in terms of downstream classification accuracy, than the classic approach which trains a discriminator and a generator. The authors then discuss potential limitations such as scalability to harder datasets and generality to other tasks.

**Questions:**

Please see the Weakness.

**Strengths And Weaknesses:**

**Strengths:**

1.This paper is very well written. Contributions and potential limitations are discussed adequately.

2.This paper improves the SOTA data utility by a large margin of differentially private data generation. The improvement is in terms of the downstream classification accuracy.

**Weakness:**

1.The technique contribution is limited. This paper mainly uses two existing techniques, DP-SGD and gradient matching.

2.This work only uses one algorithm from data condensation, i.e., gradient matching. It would be better if the authors can try more algorithms so the community can have a better understanding about data condensation for differentially private data generation. For example, distribution matching [1] that minimizes the distance between the averaged feature of real data and the averaged feature of synthetic data, which is also easy to implement with DP.

3.In Section 6 you show the generator from a previous work can improve the visual quality of your algorithm. How does the visual quality of your algorithm compare with the visual quality of data directly generated from that generator?

4.(Minor) Line 121, minimized -> minimize.

[1]: DATASET CONDENSATION WITH DISTRIBUTION MATCHING, https://arxiv.org/pdf/2110.04181v1.pdf.

---

> ### Author Response · Authors · 2022-08-02
> **Response to Reviewer qnNn**
>
> We thank all reviewers for providing valuable and constructive feedback. We are encouraged that all reviewers have given positive comments, finding our work **"presents an interesting novel perspective"** and **"could have a large impact"**, the paper is **"very well written"** and **"easy to follow and well organized"**, the experimental evaluation is **"extensive"** and **"improves the SOTA data utility by a large margin"**.
>
> We now address individual concerns of **Reviewer qnNn**. All minor points are incorporated in the revised manuscript, where the modified text is marked in **blue**.
>
> 1. **[It would be better if the authors can try more algorithms so the community can have a better understanding about data condensation for differentially private data generation. ]**
> We agree that exploring different dataset distillation approaches in the scenario of DP data generation would be of great value to the community. And we provide a discussion regarding this in the revised supplementary material (Section D.1). However, as we discussed in the revised supplementary material, other common dataset distillation approaches are generally less amenable to DP training: For example, [1] (the paper suggested by the reviewer) heavily relies on the per-class sampling of data, which contradicts label privacy and the requirement of uniform sampling for the privacy cost computation (See supplementary material for more details).
> &nbsp;
>
> 2. **[How does the visual quality of your algorithm compare with the visual quality of data directly generated from that generator?]**
> Please note that we are only using DC-GAN **architecture** as a prior, and are **not** using any pre-trained weights. Sampling from a DC-GAN generator (without training by our method) is equivalent to sampling from a randomly initialized network, which will yield basically random noise as outputs.
> &nbsp;
>
> [1]: DATASET CONDENSATION WITH DISTRIBUTION MATCHING

---

> > ### Comment · Reviewer_qnNn · 2022-08-05
> > **Thanks for the response**
> >
> > Thank you for providing the new discussion. I have no further questions.

---

### Author Response · Authors · 2022-08-02
**General Response to All Reviewers**

We thank all reviewers for providing valuable and constructive feedback. We are encouraged that all reviewers have given positive comments, finding our work **" presents an interesting novel perspective"** and **“could have a large impact”**, the paper is **"very well written"** and **"easy to follow and well organized"**, the experimental evaluation is **"extensive"** and **"improves the SOTA data utility by a large margin"**. We would be happy to answer any further questions.

---

> ### Author Response · Authors · 2022-08-02
> **Concerning Recent Relevant Publication**
>
> Concerning recent relevant publication [1]: while this paper is published and has been made public after our submission (hence concurrent work) and it did not come up during the reviews, we feel urged to make a short statement, as it looks relevant and also received an award at ICML'22. This particular approach is relying on very strong assumptions and also the derivation has turned out to be problematic. This has been openly discussed in the community since then. We would appreciate advice from the committee on if and how to incorporate this reference in a final version. We would be happy to include a reference like "Recent work [1] has shown promising results in dataset distillation under privacy concerns, but obtaining strict privacy guarantees has remained challenging.", but appreciate suggestions.
>
> [1] “Privacy for Free: How does Dataset Condensation Help Privacy?”, ICML 2022

---

### Public Comment · ~Dionysis_Manousakas1 · 2023-02-10
**Related work**

This paper from NeurIPS 2020 https://proceedings.neurips.cc/paper/2020/hash/ab452534c5ce28c4fbb0e102d4a4fb2e-Abstract.html proposes a similarly motivated and implemented approach for Bayesian learning using private pseudo data.

---

### Meta-Review · Area_Chair_R4yn · 2022-08-23

**Recommendation:** Accept
**Confidence:** Certain

**Metareview:**

The paper received overall positive reviews. The rebuttal clarified all the concerns the reviewers had. At this point, I will recommend the authors to incorporate the comments from the discussion into the paper, especially the new results with CIFAR-10.

**Award:**

No

---

### Decision · Program_Chairs · 2022-09-14

Accept